# Structure, Activity, and Function of the Protein Lysine Methyltransferase G9a

**DOI:** 10.3390/life11101082

**Published:** 2021-10-14

**Authors:** Coralie Poulard, Lara M. Noureddine, Ludivine Pruvost, Muriel Le Romancer

**Affiliations:** 1Cancer Research Cancer of Lyon, Université de Lyon, F-69000 Lyon, France; noureddinelara@gmail.com (L.M.N.); ludivine.pruvost@etu.univ-lyon1.fr (L.P.); Muriel.LEROMANCER-CHERIFI@lyon.unicancer.fr (M.L.R.); 2Inserm U1052, Centre de Recherche en Cancérologie de Lyon, F-69000 Lyon, France; 3CNRS UMR5286, Centre de Recherche en Cancérologie de Lyon, F-69000 Lyon, France; 4Laboratory of Cancer Biology and Molecular Immunology, Faculty of Sciences, Lebanese University, Hadat-Beirut 90565, Lebanon

**Keywords:** G9a, GLP, H3K9 methylation, protein lysine methylation, EHMT2, EHMT1, protein post-translational modification, cancer

## Abstract

G9a is a lysine methyltransferase catalyzing the majority of histone H3 mono- and dimethylation at Lys-9 (H3K9), responsible for transcriptional repression events in euchromatin. G9a has been shown to methylate various lysine residues of non-histone proteins and acts as a coactivator for several transcription factors. This review will provide an overview of the structural features of G9a and its paralog called G9a-like protein (GLP), explore the biochemical features of G9a, and describe its post-translational modifications and the specific inhibitors available to target its catalytic activity. Aside from its role on histone substrates, the review will highlight some non-histone targets of G9a, in order gain insight into their role in specific cellular mechanisms. Indeed, G9a was largely described to be involved in embryonic development, hypoxia, and DNA repair. Finally, the involvement of G9a in cancer biology will be presented.

## 1. Introduction

Protein lysine methylation is a dynamic post-translational modification (PTM) regulating protein stability and function. Lysine methylation of histone proteins can modulate transcriptional activity without affecting the DNA sequence itself, enabling dynamic gene transcription patterns in response to environmental stimuli [1]. Lysine methylation is deposited by writer enzymes called protein lysine methyltransferases (PKMTs), removed by eraser enzymes called lysine demethylases (PKDMs) and interpreted by reader proteins that bind to lysine methylation marks. PKMTs catalyze the transfer of the methyl group from the S-adenosyl-l-methionine (AdoMet) donor to the ε-nitrogen of a lysine residue on protein substrates [1]. The lysine ε-amino group of proteins can accept up to three methyl groups, resulting in either mono-, di-, or trimethyl lysines. To date, more than 50 PKMTs have been reported, with sequence and product specificity. Two PKMT families have been identified: the SET lysine methyltransferases containing the majority of PKMTs [2] and the Seven β-strand methyltransferase (7βS) or class I family [3]. Histones are methylated on several lysine residues. A growing number of reports also describe the methylation of non-histone proteins on lysine residues [1]. 

G9a was identified and sequenced in the 1990s [4]. It belongs to the SET PKMT family. G9a was extensively studied as a key enzyme in the mono- and dimethylation of lysine 9 of histone H3 (H3K9me1 and H3K9me2, respectively) in euchromatin [5]. Since the H3K9me2 mark is associated with transcriptional repression, G9a was primarily considered to be an epigenetic repressor [5,6,7]. Its role as a coactivator of several transcription factors emerged more recently [8,9,10,11,12]. Though G9a is the most commonly used term for this lysine methyltransferase, it is also known as lysine methyltransferase-1C (KMT1C), euchromatic histone N-methyltransferase 2 (EHMT2), or BAT8 (HLA-B associated transcript 8).

The current review will provide an overview of the structural features of the protein with a particular focus on its paralog GLP (G9a-like protein). The biochemical features of G9a will also be detailed with a special emphasis on the key PTMs affecting G9a and regulating its activity and function. Finally, among the large number of G9a substrates described, including histone and non-histone substrates, the present report will focus on their involvement in specific physiological pathways and their connection to cancer. 

## 2. Structural Features

### 2.1. Structure and Domain Architecture

In human cells, G9a exists as two isoforms: a full-length isoform of 1210 amino acids (called isoform A) derived from 24 exons of the G9a gene and a splice variant of 1176 amino acids (isoform B) that arises from the excision of exon 10 (Figure 1a). The alternative splicing of G9a is conserved in different species, tissues, and cell lines [13]. Even if the two isoforms are ubiquitously found in different tissues, the ratio between them varies. For example, isoform A is preponderant in the kidney, thymus, and testis, and, interestingly, is more abundant in epithelial cell lines compared to mesenchymal cell lines and more transformed cell lines [13]. Mauger et al. reported that the two isoforms display similar methyltransferase activities and subcellular localizations. Likewise, Fiszbein et al. showed that isoform B expression increased during neuronal differentiation [14]. They did not report any change in G9a catalytic activity following exon 10 inclusion, but demonstrated that exon 10 inclusion increases G9a nuclear localization in a neuronal cell line [14]. Mouse G9a is also subjected to alternative splicing. Full-length mouse G9a protein contains 1263 amino acids and shares more than 90% homology with human G9a [15]. 

G9a belongs to the Su(var)3-9 family of methyltransferases, which was first identified in Drosophila melanogaster [16]. The main characteristic of this family of proteins is the presence of a highly conserved SET domain [17]. SET, an acronym for Su(var)3-9, Enhancer-of-zeste and Trithorax, is a long sequence of 130 to 140 amino acids, characterized in 1998, that has a unique structural fold [17]. The SET domain is composed of a series of β strands that fold into three sheets and surround a knot-like structure [18]. The conserved core of the SET domain is flanked by a pre-SET (nSET) domain providing structural stability by interacting with different surfaces of the core SET domain, and a post-SET (cSET) domain responsible of the formation of a hydrophobic channel via an aromatic residue [19]. Neither pre-SET nor post-SET domains are conserved across KTM SET domains, as they vary in size and tertiary structure [20]. In the core SET domain, G9a contains an inserted i-SET domain (Figure 1a). The i-SET domain forms a rigid docking platform and a substrate binding groove with the post-SET domain in three-dimensional structures [21]. The G9a SET domain contains four structural zinc fingers for proper folding and enzymatic activity. A cluster of three Zn^2+^ ions is chelated by nine cysteines, whereas the fourth Zn^2+^ ion, adjacent to the S-adenosylmethionine (SAM)-binding site, is chelated by four cysteines [22]. The binding of AdoMet and the protein substrate occurs on opposite sides of the SET domain. AdoMet binds and positions its methyl group at the base of the channel, while the side chain of the target lysine protrudes into the channel [20]. Within the SET domain, the tyrosine residue Y1154 was demonstrated to be essential for the catalytic activity of G9a [23]. The tyrosine may allow deprotonation of the positively charged ammonium group in order to favor methylation. 

G9a also contains a cysteine-rich region, a polyglutamate region and seven ankyrin repeats of 33 amino acids (Figure 1a). The ankyrin repeat domain was reported to be a mono- and dimethyllysine binding module, a reader domain important for protein-protein interactions [24]. The specificity of the G9a ankyrin repeat domain is comparable to the specificity of other groups of reader proteins recognizing methyl binding protein modules, such as the chromodomain, the tudor domain, or the PHD finger domain [24]. G9a was the first protein described to harbor within a single polypeptide, the signal to catalyze and read the same epigenetic marks, H3K9me1, and H3K9me2 [24]. 

A nuclear localization signal was identified in the N-terminal region of human G9a [25], and amino acids 1-280 of human G9a were shown to act as a coactivator domain in transient reporter gene assays [10] (Figure 1a).

### 2.2. GLP, a G9a Paralog

A paralog of G9a was identified and called G9a-like protein (GLP), though it is also termed lysine methyltransferase-1D (KMT1D) or euchromatic histone N-methyltransferase 1 (EHMT1) (Figure 1b). G9a and GLP share 45% sequence identity and around 70% sequence similarity (Figure 1c) [2]. They differ primarily in the N-terminus, and present a high level of conservation in the SET domain with over 80% shared sequence identity (Figure 1c) [27]. The main difference in structure between the two proteins concerns the E-rich domain of G9a, which is composed of a sequence of repeated glutamic and aspartic acid residues in the case of GLP (Figure 1b, c). In addition, binding affinities of the ankyrin domains of G9a and GLP for H3K9 differ, as GLP and G9a preferentially bind to mono- and dimethylated H3K9, respectively [24,28].

G9a and GLP form homo- and heterodimers via their SET domains in complex with ZNF644 and WIZ [6,29,30,31]. In the endogenous complex, they act mainly as heterodimers in a large variety of human cells [6]. However, in vitro, independently of each other, G9a and GLP are able to catalyze lysine methylation by forming homodimers. Extensive research has focused on G9a, albeit GLP seems to be equally important for most biological phenomena ascribed to G9a. Indeed, GLP generally possesses similar catalytic activities as G9a [29]. However, the individual effects of G9a and GLP are hard to study, as G9a depletion destabilizes GLP [6,32]. 

## 3. Biochemical Features

### 3.1. Sequence Specificity

The majority of studies conducted on G9a sequence specificity focused on Histone H3. In vitro, the minimum substrate recognition site of seven amino acids of H3 is composed of residues 6 to 11 (TARKSTG), with a consensus methylation site encompassing RK/ARK [33]. The arginine residue adjacent to the lysine residue is essential for G9a activity [33]. G9a preferentially acts when a hydrophobic amino acid is positioned before the arginine residue, such as alanine. After the lysine residue, G9a favors a hydrophilic residue followed by a hydrophobic one. This G9a recognition site is present in several non-histone proteins, as well as on its N-terminal domain [34,35,36]. 

Several biochemical studies have shown that specific PTMs affect the catalytic activity of G9a. For instance, phosphorylation of S10 or T11 of H3 impairs G9a catalytic efficacy [33,36]. In addition, R8 of H3 can be methylated by the arginine methyltransferase PRMT5 in vivo, and this event impairs methylation of H3K9 by G9a [36]. Indeed, a decrease in methylation of over 80% was reported for peptides carrying an asymmetric dimethylation of R8, a methylation mark catalyzed by PRMT5 [36].

### 3.2. Product Specificity

G9a mainly catalyzes mono- and dimethylation events, as illustrated with H3K9 [6,24]. However, several reports demonstrated that G9a also generates, after a long incubation time, trimethylation of H3K9 (H3K9me3) [25,37]. Investigations on G9a-deficient cells demonstrated that G9a is the major H3K9me1 and H3K9me2 methyltransferase of euchromatin [5].

Biochemically, the specificity of G9a methylation for a particular state is largely due to a tyrosine residue in its active site. Indeed, Y1067 controls whether G9a catalyzes mono-, di- or trimethylation of lysines; Y1067 mutation to F1067 allowing G9a trimethylation of H3K9 [21]. Mechanistically, Y1067 forms hydrogen bonds with the nitrogen atom of the ε-amino group of the target lysine residue [21].

### 3.3. Regulation

#### 3.3.1. PTMs 

As for most proteins, G9a is subjected by many PTMs that regulate its ability to bind new partners and impact its cellular functions (Figure 2). Further details about their cellular features will be given in the corresponding sections below. 

G9a was shown to be auto-methylated on lysine 185 (K185) and phosphorylated by the Aurora kinase B (AurKB) on the adjacent threonine 186 (T186) in the N-terminal domain of the protein [35] (Figure 2). Heterochromatin protein 1 proteins (HP1α, HP1β, HP1γ) and CDYL (chromodomain Y-like) were identified as specific partners that bind methylated G9a [34,35]. These proteins contain chromodomains functioning as methyl-lysine binding modules. Of note, a similar methylation and phosphorylation switch on adjacent residues was previously demonstrated for the histone H3 [38,39]. H3K9me2 methylated by G9a recruits HP1 proteins, whereas H3 phosphorylated on S10 by AurKB has an opposite effect [38,39]. Like G9a, GLP is also auto-methylated on lysine 205 (K205) and phosphorylated by AurKB on threonine 206 (T206) [32]. Both G9a and GLP auto-methylation sites can be demethylated by the KDM4 lysine demethylase family [40]. Sampath et al. found no evidence of a role for G9a auto-methylation in the regulation of G9a enzymatic activity [35]. 

Additionally, G9a was shown to be phosphorylated on two serine residues involved in DNA damage repair, namely Serine 211 (S211) phosphorylated by casein kinase 2 (CK2) and serine 569 (S569) phosphorylated by ATM kinase (Figure 2) [41,42]. Interestingly, phosphorylation of G9a on S211 does not change its methyltransferase activity and G9a catalytic inhibitor does not affect G9a phosphorylation on S569 [41,42].

G9a is sumoylated in skeletal myoblasts in order to regulate its transcriptional activity [43]. This event acts as a signal for the recruitment of the histone acetyltransferase PCAF (p300/CBP-associated factor) to E2F1 target genes, implicated in cell cycle progression by increasing the level of histone H3 lysine 9 acetylation [43].

Casciello et al. demonstrated that G9a stability is regulated by proline hydroxylation catalyzed by oxygen sensors, as inhibition of the latter increased protein stability [44]. Authors showed that G9a hydroxylation is detected in normoxic conditions, whereas it is not detected under hypoxia. Proline hydroxylation occurs on proline residues 676 (P676) and 1207 (P1207) in consensus hydroxylation motifs LXXLAP and leads to efficient degradation by the proteasome (Figure 2) [44]. G9a is also hydroxylated in the ankyrin repeat domain of G9a on asparagine 779 (N779) by the asparaginyl hydroxylase factor inhibiting HIF (FIH) (Figure 2) [45]. This event impedes G9a binding to methylated H3K9 products and to di- and trimethylated H3K9. Hydroxylation of N779 destabilizes the interaction of H3K9me2 with the ankyrin repeat domain of G9a by disrupting the structural pocket that facilitates methyl binding [24,45]. Likewise, GLP is hydroxylated on N867 [45]. 

#### 3.3.2. Stability 

G9a protein stability relies on the presence of GLP, as GLP depletion also decreases G9a expression [6,32]. Using G9a^−/−^ and GLP^−/−^ embryonic stem cells, Tachibana et al. reported that G9a is more stable in the G9a/GLP heteromeric complex. This observation did not apply to GLP [6]. The protein WIZ was reported to be a key partner of both G9a and GLP to stabilize the G9a/GLP heteromeric complex [30]. Both WIZ and GLP depletion decreases G9a protein levels, suggesting that the WIZ/G9a/GLP complex protects G9a from degradation [30]. Later, Bian et al. mapped the specific sequence of WIZ interacting with G9a/GLP. They showed that WIZ only interacts directly with the NTD of GLP [31]. Its interaction with G9a might be indirect and mediated by the fact that G9a and GLP form heterodimers. WIZ contains multiple zinc finger motifs, targeting the G9a/GLP complex to chromatin in order to mediate H3K9 methylation [31].

### 3.4. Substrates

#### 3.4.1. Histone Substrates

In 2001, Tachibana et al. identified the first substrates of G9a as histone proteins [46] (Table 1). They demonstrated that G9a was able to add methyl groups to H3 on lysine 9 and lysine 27 [46]. Since then, G9a has largely been described as the major PKMT catalyzing the mono- and dimethylation of H3K9 [5], and, to a lesser extent, H3K9 trimethylation [25,37]. Though H3K9 methylation is well known for its role in transcriptional silencing [6,47], the impact of H3K27 methylation by G9a emerged more recently. Wu et al. demonstrated in 2011 that even though H3K27me2/3 is not affected in G9a^-/-^ ES cells, H3K27me1 levels were clearly lower in these cells [48]. G9a also methylates H3 on lysine 56 (H3K56me1) in order to maintain proper DNA replication [49], a methylation event that was shown to be induce by DNA damage [41]. 

G9a methylates histone H1 in a variant-specific manner. Human cells have 11 H1 variants, two of which were shown to be methylated by G9a, namely isotype 2 (H1.2) and isotype 4 (H1.4) [50,51]. H1.4 was reported to be mono- and dimethylated on H1.4K26. This event provides a recognition site for HP1 binding, establishing a proper chromatin surface and suggesting a role for H1.4K26me1/2 in transcriptional repression [50]. G9a methylates H1.2 on K187 in vitro and in vivo. However, H1.2K187me2 is not recognized by HP1 proteins, demonstrating selective recognition by these proteins [51]. Weiss et al. demonstrated that G9a does not directly bind to methylated histone variants, suggesting a different mechanism from that observed in H3K9me1/2 to achieve methylation [51].

#### 3.4.2. Non-Histone Substrates

G9a also methylates a large number of non-histone proteins involved in several biological functions listed in Table 2. Most of these are linked with transcriptional regulation, as G9a methylates numerous transcription factors, chromatin remodeling factors, and coregulators. 

### 3.5. Inhibitors

Among the numerous G9a inhibitors, there are three different types: (i) substrate competitive inhibitors, (ii) SAM cofactor competitive inhibitors and (iii) inhibitors by ejection of Zn^2+^ ions. Substrate competitive inhibitors act by binding to G9a substrate binding sites, while SAM inhibitors prevent G9a-mediated methylation by interacting with SAM binding sites on G9a [52]. Most of these inhibitors also impact GLP [53].

#### 3.5.1. Substrate Competitive Inhibitors

Substrate competitive inhibitors specifically bind to the substrate binding site of G9a. The first substrate competitive inhibitor discovered was BIX01294, a quinazolin derivative able to inhibit H3K9me2 [70]. Many studies then sought to optimize this inhibitor by enhancing its G9a specificity, efficacy and by reducing cell toxicity. Based on Structure-Activity Relationship studies (SAR), modifications of BIX01294 provided more specific and powerful G9a inhibitors including UNC0224, UNC0321, UNC0638, UNC0646 [52]. The majority of G9a substrate competitive inhibitors impede G9a activity by interacting with two G9a aspartate residues in the SET domain (D1074 and D1083) [71,72]. Recently, by adding and expanding the 1,4 benzodiazepine cycle, Milite et al. improved UNC0638 potency and named it EML741 [73]. 

#### 3.5.2. SAM Competitive Inhibitors

The cofactor SAM is the methyl donor essential for G9a-mediated methylation. SAM competitive inhibitors compete with SAM to bind to the SAM binding site of G9a. The first inhibitor of this class to be identified by Kubicek et al. was BIX01338, discovered around the same time as BIX01294 [70]. Analogous inhibitors were then synthetized with similar structures, such as BRD9536 and BRD4770 [74]. However, this type of inhibitor remains less specific than substrate competitive inhibitors, as it also downregulates the enzymatic activity of several other PKMTs [52]. 

#### 3.5.3. Inhibition by Ejection of Structural Zn^2+^


Lastly, Lenstra et al. reported that structural zinc ions are essential to maintain the enzymatic activity of the methyltransferases G9a/GLP [22]. By using selenium- or sulfur-containing proteins able to eject the fourth structural zinc ions, they demonstrated that G9a methyltransferase activity could be inhibited. Molecules used clinically such as ebselen, disulfiram, and cisplatin work specifically as inhibitors of G9a and GLP. These findings may offer new perspectives to develop further G9a-specific inhibitors [22].

## 4. Cellular Features

### 4.1. Connection with Chromatin Regulation

#### 4.1.1. G9a Corepressor Functions

As mentioned above, G9a is a coregulator with an essential role in repression of gene transcription. Functionally, G9a is involved in several mechanisms, primarily the methylation of the histone H3 N-terminal tail in order to close chromatin (Table 1).

G9a in Euchromatin

Numerous studies have shown that G9a is recruited to specific target genes as a corepressor by transcription factors, such as CCAAT displacement protein/cut (CDP/cut) [75], growth factor independent 1 (Gfi1) [76], positive regulatory domain I-binding factor 1 (PRDI-BF1) [77], neuron restrictive silencing factor (NRSF) (also known as REST) [78], multi-domain protein UHRF1 [79], and the noncoding RNA Air [80], in order to remodel chromatin structure. G9a also represses active gene transcription by recruiting other corepressors. For example, in euchromatin, G9a interacts with Polycomb Repressive Complex 2-proteins, including the PKMT EZH2, in order to transcriptionally silence specific regions within the genome (Figure 3a) [81]. 

G9a in heterochromatin

In heterochromatin, G9a drives silencing mechanisms by serving as a platform for the formation of repressive complexes. Methylation of H3K9 leads to the recruitment of proteins such as HP1, which can bind to methylated H3K9 via their chromodomains [38,39]. This recruitment is crucial for heterochromatin formation and gene silencing [82]. In addition, G9a also recognizes H3K9 methylation via its ankyrin repeat in order to work as a scaffold for the recruitment of other corepressors [24]. It was shown for instance that G9a interacts with the PKMT Suv39h and SETDB1 in specific regions of heterochromatin to maintain chromosomal stability (Figure 3a) [83].

G9a and DNA methylation

Other mechanisms underlying G9a repressive function have been identified. For example, the ankyrin repeat domain of G9a was reported to contribute to DNA methylation-mediated repression of transcription by recruiting DNA methyltransferases (DNMT3a and DNMT3b), and by recognizing the H3K9me2 histone mark [24,84]. A specific residue of the ankyrin repeat domain (Asp905) has also been associated with this co-repressive function by maintaining H3K9me2 levels and establishing DNA methylation [85]. In addition, Chang et al. demonstrated that G9a dimethylates DNMT3a on K47, allowing its recognition by the MPP8 chromodomain [66]. This event results in a silencing complex containing DNMT3a/MPP8/G9a on chromatin that could in part explain the co-occurrence of DNA methylation and H3K9 methylation in chromatin (Figure 3a). Additionally, Smallwood et al reported that HP1 proteins, the readers of H3K9 methylation, target DNMT1 enzyme to euchromatic sites, providing a basis for the generation of CpG methylation [86]. Finally, DNMT1 is methylated by G9a reinforcing the whole model [36] (Figure 3a). 

#### 4.1.2. G9a Coactivator Functions

In addition to the well-studied and established co-repressive function of G9a, reports have emerged on its function as a coactivator, by contributing to the activation of gene expression [9,10,11,12,32,87,88]. 

It was suggested that different binding partners may play critical roles in the switch between the coactivator and corepressor functions of G9a. Indeed, G9a stabilizes the occupancy of the Mediator complex on the promoter of the adult β globin gene in a NF-E2/p45-dependent manner to exert its coactivator function, while it recruits the H3K4 demethylase Jarid1a to the promoter of the embryonic β globin gene and results in transcription repression [12,89] (Figure 3b). It has also been shown that G9a is recruited to the promoter or enhancer regions of its positively regulated target genes, indicating that G9a may act directly on their expression [8,9,10,11,12,32,87,88,89]. In addition, G9a was reported to bind to RNA polymerase II, indicating that G9a may be involved in the establishment of a preinitiation or initiation complex during transcription [12].

The G9a activation domain (AD) (amino acid 1–280 in human G9a) was first identified by Dr. Stallcup’s group using transient reporter gene assay [10] (Figure 1). G9a AD is sufficient and required for its coactivator function [10] and contains an autonomous activation domain [9]. Recently, we demonstrated the importance of G9a auto-methylation in the G9a AD for its coactivator function. Indeed, auto-methylation of G9a (K185) is required for its coactivator function with the glucocorticoid receptor (GR), by facilitating the binding of HP1γ and the subsequent recruitment of RNA pol II [32]. Inversely, G9a phosphorylation (T186) by AurKB antagonizes these effects (Figure 3b). Thus, these adjacent modifications regulate coactivator functions and contribute to determining whether G9a act as a coactivator or corepressor [32]. At the physiological level, we demonstrated that the coactivator activity of G9a regulates migration of the lung cancer cell line, A549 [32], and GC-induced cell death in leukemia [32,88]. In addition, G9a was reported to function as a scaffold protein to recruit the coactivators p300 and CARM1 on a subset of GR target genes, leading to transcriptional activation [8,9]. 

G9a also acts as a coactivator by specifically methylating the estrogen receptor alpha (ERα) on K235 [58]. This event is recognized by the Tudor domain of PHF20, which recruits the MOF histone acetyltransferase complex in order to acetylate H4K16 and promote active transcription (Figure 3b. Through this mechanism, G9a regulates a specific subset of ERα target genes [58]. 

### 4.2. Cellular Roles and Functions

#### 4.2.1. Embryonic Development

Most PKMTs are essential for the formation of healthy embryo, as they remodel histones and control chromatin packaging and transcriptional accessibility along the genome [1]. Hence, it came as no surprise that G9a knockout impacted embryonic development [5]. Embryo of mice genetically engineered to be G9a-deficient displayed delayed development, growth arrest by the earliest stages monitored, and were no longer viable by embryonic day 9.5 [5]. Histones extracted from G9a-deficient embryos showed a strong decrease in H3K9me2 [5,6] Later studies, then reported the importance of G9a in specific developing tissues and organs based on different analyses. 

Germ Cell Development

Germ line-specific G9a knockout mice were shown to be sterile due to a drastic loss of mature gametes [90]. In addition, completion of meiosis was not observed in either gender. In G9a-deficient germ cells, H3K9me1/2 decreased during meiosis, suggesting that gene silencing induced by G9a is crucial for proper meiotic prophase progression [90]. 

Cardiac Development

Engineered mice in which GLP was knocked out and G9a knocked down in cardiomyocytes showed neonatal lethality and atrioventricular septal defects, strongly implicating G9a and GLP in cardiomyocyte function for atrioventricular septum formation [91]. However, cardiomyocyte-specific G9a knockout mice were normal and the loss of G9a induced only a slight decrease in H3K9me2 levels in cardiomyocytes, indicating that adequate H3K9me2 can be performed by enzymes other than G9a in cardiomyocytes [91]. 

Neuronal Development

Neuron-specific deficiency of G9a did not reveal obvious neuronal developmental or architectural defects [92]. However, these mice displayed various abnormal phenotypes, including defects in cognition and adaptive behaviors, such as difficulties in learning, motivation and environmental adaptation [92]. Authors demonstrated that multiple non-adult neuronal and non-neuronal progenitor genes were derepressed in the forebrain of these mice deficient for G9a [92]. Using pharmacological inhibition of G9a/GLP activity, it was demonstrated that G9a/GLP are required in the dorsal hippocampus for the transcriptional switch from short-term to long-term spatial memory formation [93]. Repression of G9a and H3K9 methylation has been described in postmortem nucleus accumbens of human cocaine addicts, indicating a clinical relevance of G9a in human addiction [94]. Through extended analyses, Maze et al. demonstrated a role for G9a in neuronal subtype identity in the adult central nervous system, and a critical function for G9a and H3K9 methylation in the regulation of behavioral responses to environmental stimuli [95]. 

Bone Formation

G9a protein levels and H3K9me2 were reported to increase during developmental progression in tooth and growth plate cartilage [96]. G9a methyltransferase activity regulates cell proliferation and differentiation in dental mesenchyme in order to promote proper tooth development [96]. 

Using two different models of conditional G9a knockout mice, G9a was shown to be involved in cranial bone formation, since mutant mice had severe defects in cranial vault bones with opened fontanelles [97,98]. Mechanistically, the effect of G9a on cranial bone formation relies on its function as repressor of Twist expression during osteoblastic differentiation and as coactivator of RunX2 [97,98]. Stallcup’s group demonstrated that G9a is able to enhance RunX2-mediated transcription in transient reporter gene assays by acting as a coactivator of RunX2 [11]. RunX2 is a key transcription factor of bone-forming cells by regulating osteoblastic differentiation [99]. Later, Ideno et al. showed that G9a enhances RunX2 transcriptional activity in mesodermal cells through binding and activation of RunX2 [97]. 

Other Mechanisms

G9a knockdown or inhibition through pharmacological inhibitors in adult erythroid cells induces re-emergence of a fetal gene program, illustrated by the switch in expression from adult to fetal β-globin isoforms [12,89] (Figure 3).

Conditional knockout of G9a in the skeletal muscle lineage highlighted that G9a has little effect on skeletal myogenesis [100].

Targeted depletion of G9a in the developing mouse retina generated disorganized tissues [101]. According to the authors this was due to the fact that retinal progenitor cells depleted for G9a were highly proliferative and were not able to mature into the specialized components of the retina [101]. Similar results were obtained in zebrafish embryos knocked down for G9a using morpholino antisense oligos [102]. 

These different studies clearly demonstrated that G9a has a major impact on embryonic development, with roles in various pathologies, including neurological disorders, cardiac pathogenesis, immune cell development, and cancer progression. 

#### 4.2.2. Hypoxia

In mammalian cell lines, G9a activity was reported to increase under hypoxic conditions, concomitant to an increase in total H3K9me2 levels, resulting in gene silencing [103]. In G9a^-/-^ mouse embryonic stem cells under hypoxic conditions, the level of H3K9me2 was significantly lower, demonstrating that G9a was involved in hypoxia-induced H3K9me2 [103]. The hypoxic upregulation of G9a was attributed to specific PTMs (Figure 4). As described previously, G9a is hydroxylated at residues P676 and P1207 by PHD1 in order to target G9a toward proteasome degradation via ubiquitinylation [44]. Hypoxia induces PHD1 inhibition and a subsequent upregulation of G9a, leading to an increase in H3K9me2 and the silencing of a specific subset of target genes. Casciello et al demonstrated that G9a inhibition decreases proliferation, migration, and in vivo tumor growth [44]. Likewise, in ovarian cancer, FIH reaction was limited under hypoxia, leading to a reduced expression of metastasis-suppressor genes via H3K9 methylation [45]. Mechanistically, FIH induces hydroxylation of G9a on N779, impairing its ability to bind mono- and dimethylated H3K9, and thus methylate H3K9 [45] (Figure 4). 

However, the role of G9a under hypoxia is likely more extensive, as G9a methylates many protein substrates involved in hypoxia, namely Pontin, Reptin, and HIF-1α [61,62,63] (Figure 4). Bao et al. demonstrated that HIF-1α, a master regulator of the hypoxic response, is mono- and dimethylated by G9a on K674 [63]. They demonstrated that G9a is able to methylate HIF-1α in an oxygen-independent manner. However, endogenous HIF-1α is unstable and degraded under normoxic conditions, indicating that HIF-1α is unlikely to be methylated in normoxia [63]. HIF-1αK674me1/2 suppresses HIF-1α transcriptional activity under hypoxia and expression of its downstream target genes (Figure 4). These authors also demonstrated that HIFα methylation by G9a decreases HIF-1-dependent migration of glioblastoma cells [63]. In addition, G9a methylates Reptin and Pontin, two chromatin remodelers involved in hypoxia, known to bind to HIF-1 proteins [61,62]. Under hypoxia, G9a monomethylates Reptin on K67 (K67me1), this methylation negatively regulates a subset of hypoxia target genes via the recruitment of Reptin K67me1 to their promoters and an enhanced binding to HIF-1α [62]. In addition, Reptin K67me1 leads to the recruitment of corepressors such as HDAC1 to hypoxia-responsive gene promoters in order to decrease HIF-1α transcriptional activity [62] (Figure 4). Conversely, under hypoxia, G9a methylates Pontin on six lysine residues (K265, K267, K268, K274, K281, K285), enhancing p300 coactivator recruitment on the promoters of HIF-1α target genes, resulting in an increase in HIF-1 transcriptional activity [62] (Figure 4). Although Reptin and Pontin share similarities in their structures, they act as coactivator or corepressor of HIF-1 depending on their subset of target genes in order to modulate cellular responses to hypoxia [61,62]. 

The ability of G9a to repress genes under hypoxic conditions suggests a key role for G9a in cell survival processes in this condition, especially in solid tumors where hypoxia is a common microenvironmental state.

#### 4.2.3. DNA Damage and DNA Repair

Two reports demonstrated that G9a was recruited to DNA-damage sites, mainly through G9a phosphorylation [41,42]. G9a is phosphorylated by casein kinase 2 (CK2) at S211 in response to DNA double-strand breaks (DSBs), promoting G9a recruitment to sites of DNA damage by increasing its interaction with chromatin, where it can directly interact with replication protein A (RPA) [42]. In turn, binding of G9a to RPA modulates RPA and Rad51 foci formation, allowing efficient homologous recombination of DSBs and cell survival [42]. In parallel, Ginjala et al. demonstrated that G9a is phosphorylated by ATM kinase on S569 [41]. This event also leads to its recruitment to sites of DNA breaks. Authors demonstrated that the catalytic activity of G9a is critical for early recruitment of 53BP1 and BRCA1 to DNA lesions, but dispensable for their late recruitment. Induction of DSBs leads to an increase in H3K9me2 and H3K56me1 in their neighboring chromatin, two histone targets of G9a [41]. Inhibition of the catalytic activity of G9a decreases these modifications, suggesting that G9a could be recruited to DNA breaks in order to induce local histone methylation and subsequent local transcriptional silencing. Finally, using GFP-based reporters of homologous repair (HR) or non-homologous end-joining repair (NHEJ), they demonstrated that the catalytic activity of G9a impairs both mechanisms, HR and NHEJ [41]. Moreover, phosphorylation of S211 and S569 appears to be essential for proper DNA repair [41,42]. 

G9a may also methylate specific non-histone proteins involved in DNA repair mechanisms, such as Polo-loke kinase 1 (Plk1) and p53 [57,68]. Plk1 phosphorylation on T210 is required during DNA damage repair and checkpoint recovery [104]. Recently Li et al. demonstrated that the activity of Plk1 is controlled by a switch between methylation and phosphorylation, as for G9a and GLP [68]. Authors showed that under DNA damage stress conditions, the interaction between G9a and Plk1 is enhanced and G9a monomethylation on K209 of Plk1 is increased [68]. Interestingly, Plk1 methylation by G9a is not necessary for its recruitment to DNA lesions or for the assembly of the DNA repair machinery via RPA and Rad51 recruitment. However, this methylation is crucial for the timely removal of this DNA repair machinery from DNA lesions, which is essential for the proper completion of DNA damage repair [68]. The tumor suppressor p53 was also demonstrated to be a substrate for G9a on K373 [57]. However, p53 methylation seems to be link with inactive p53, as the level of methylated p53 during DNA damage does not change even though the total level of p53 increases dramatically [57]. This data is consistent with the fact that catalytic inhibition of G9a using inhibitors under low DNA damage conditions impairs DNA DSB repair in a p53-independent manner [105]. However, it is interesting to note that G9a dimethylation of p53 at K373 increases Plk1 expression and promotes colorectal cancer [106].

These reports clearly demonstrate the relevance of G9a in the maintenance of genome integrity, implicating G9a in cancer biology. 

## 5. G9a in Cancer

### 5.1. G9a Oncogenic Role

Recently, dysregulations in the PTMs of both DNA and histones were shown to contribute to cancer initiation and progression [107]. These epigenetic modifications, which result in altered chromatin structure and gene expression were reported in different types of cancers [108] (Figure 3). G9a was overexpressed in breast, gastric, ovarian, cervical, endometrial, prostate, lung, colorectal, liver, urinary bladder, and brain cancers, as well as in hematological malignancies, melanoma, and cholangiocarcinoma, leading to aberrant H3K9 methylation [109,110,111,112,113,114,115,116,117,118,119,120,121,122]. One of the main reasons for this increase in G9a expression and H3K9 methylation is hypoxia [103]. The molecular mechanisms associated with this phenomenon are described in a previous section (Figure 4). Furthermore, high levels of G9a expression were associated with poor prognosis and shorter survival in cancer patients [57,123,124,125,126,127]. G9a involvement in cancer biology is likely due to its pivotal role in tumor cell proliferation, survival, and metastasis primarily by controlling several transcription programs (Table 3).

#### 5.1.1. Breast Cancer

High G9a-mediated H3K9 methylation triggers the proliferation and progression of breast cancer (Table 3) [109,128,129]. For instance, G9a overexpression was shown to down-regulate the expression of some tumor suppressor genes, such as ARNTL, CEACAM7, GATA2, HHEX, KLRG1, and OGN. Blocking G9a methyltransferase activity was sufficient to re-express these genes, and consequently inhibit breast cancer cell proliferation and migration in vitro and tumor growth in vivo [44]. G9a was also demonstrated to interact with MYC and suppress its target genes by favoring H3K9me2, in order to stimulate MYC-dependent breast tumor growth [129]. G9a may also contribute to enhancing breast tumor metastasis by silencing several genes implicated in epithelial-mesenchymal transition (EMT), namely the two anti-metastatic tumor suppressor genes, desmocollin 3 (DSC3), belonging to the cadherin superfamily, and the protease inhibitor MASPIN, which were transcriptionally reactivated in a dose-dependent manner upon inhibition of G9a activity, concomitantly to a significant decrease in global H3K9 dimethylation [130]. In addition, in EMT, G9a was shown to repress the expression of E-cadherin, a cell adhesion factor, upon association with the SNAIL transcription factor and to induce H3K9me2 of its promoter [131]. Depletion of G9a restored E-cadherin expression and inhibited breast cancer cell migration and invasion in vitro and in vivo [131]. G9a also silenced the expression of the type-II cadherin CDH10 through histone methylation, stimulating hypoxia-mediated cellular motility; and its inhibition prevented cellular movement and breast cancer cell colonization in the lungs [123]. G9a methyltransferase activity was further reported to (i) collaborate with the transcription factor YY1 and HDAC1 to disrupt cellular iron homeostasis by repressing ferroxidase hephaestin, resulting in iron accumulation and breast cancer progression [109], (ii) induce breast cancer cell autophagy by modulating the AMPK-mTOR pathways [132], and (iii) promote breast cancer recurrence through the suppression of pro-inflammatory genes [133].

#### 5.1.2. Gastric Cancer

In gastric cancer, G9a activation reduces apoptosis and promotes tumor cell growth (Table 3) [134]. For instance, blocking the catalytic activity of G9a reduces cell growth and autophagy by downregulating the mechanistic target of rapamycin (mTOR) pathways. Authors showed that G9a activates mTOR through H3K9 monomethylation at the mTOR promoter [125]. G9a inhibition by (i) kaempferol, a flavonoid present in fruits and vegetables [135], (ii) SH003, an herbal formulation [136], or (iii) cinnamaldehyde (CA), the bioactive ingredient in Cinnamomum [137], stimulated autophagic gastric cancer cell death. Increase in H3K9 methylation under hypoxia also mediated the silencing of the tumor suppressor gene, runt-domain transcription factor 3 (RUNX3) [138]. Finally, G9a overexpression was shown to upregulate the expression of ITGB3, an integrin family member, in an enzyme-independent manner inducing gastric cancer metastasis [139].

#### 5.1.3. Human Reproductive Cancers

Alterations in G9a expression were also associated with human reproductive cancers (Table 3). In ovarian cancer (OCa), high G9a expression levels were correlated with late stage, high grade, and a decreased overall survival in OCa patients [111,140]. An elevation in the level of G9a was observed in vitro in invasive cell lines ES-2, SKOV-3, TOV-21G, OV-90, and OVCAR-3, and in vivo in metastatic lesions in comparison with less aggressive tumor cells and primary tumors [111]. Depletion of G9a inhibited cellular adhesion, migration, invasion, and anoikis-resistance of OCa cell lines in vitro and suppressed OCa metastasis in vivo [111]. Further investigations revealed that several tumor suppressor genes were repressed in OCa by G9a, such as DUSP5, SPRY4, CDH1, and PPP1R15A. PARP inhibitor-resistant high-grade serous ovarian carcinoma (HGSOC) displayed an increase in H3K9me2 associated with an increase in the overall expression of G9a [140]. Similar observations were made in vivo on patient-derived xenografts, indicating that a high G9a expression maintains resistance to PARP inhibitors [140]. Interestingly, inhibition of G9a displayed synergistic anti-tumor effects in combination with DNA methylation inhibitors in OCa cell lines, where authors induced cell death by upregulating endogenous retroviruses (ERVs), consequently activating the viral immune response [141]. 

In cervical cancer, G9a induces the expression of angiogenic factors including angiogenin, interleukin-8, and C-X-C motif chemokine ligand-16, prompting angiogenesis and cancer cell invasion, and decreasing patient survival [142]. Interestingly, depletion of G9a decreased the expression of oncogenic proteins such as Bcl-2, Mcl-1, and Survivin, and increased the expression of E-cadherin inhibiting cell adhesion and invasion [112]. 

Likewise, in endometrial cancer, G9a-mediated H3K9 methylation induced tumor invasion in vitro and in vivo via the silencing of the E-cadherin [113]. Indeed, G9a depletion reduces H3K9me2 levels, restores E-cadherin expression and decreases E-cadherin promoter DNA methyltransferase recruitment. G9a expression is higher in endometrial cancer tissues and its expression is correlated with deep myometrial invasion [113]. 

Finally, in prostate cancer, high G9a expression was associated with high pathological grade and poor overall survival. In this model, G9a promoted cancer proliferation by inhibiting PI3K/AKT/mTOR pathway [114].

#### 5.1.4. Lung Cancer

In lung cancer, G9a possesses proliferative and metastatic properties (Table 3) [114]. Highly invasive lung cancer cell lines were reported to display higher G9a protein levels, in comparison with weakly invasive cells. Overexpressing G9a increased cell motility and invasiveness [143]. Different reports demonstrated that G9a induced tumor growth, invasion, and migration by (i) silencing specific EMT-regulating genes, including caspase-1 and the epithelial cell adhesion molecule Ep-CAM [124,144], (ii) mediating the Snail2-induced E-cadherin suppression [145], and/or (iii) activating the focal adhesion kinase signaling pathway [146]. Depletion of G9a abolished lung cancer cell migration and invasion in vitro and metastasis in vivo [124,144,146]. G9a also induced cell proliferation through the activation of the WNT signaling pathway by suppressing WNT signaling inhibitors like DKK1, APC2, and WIFI [121]. Moreover, G9a was shown to play an important role in maintaining lung cancer cell stemness by maintaining DNA methylation of multiple lung cancer stem cell genes and their subsequent expression [147]. 

#### 5.1.5. Colorectal Cancer

In colorectal cancer (CRC), high levels of G9a are associated with tumor initiation, maintenance, and proliferation (Table 3) [59,106,148]. In primary CRC patient samples, transcriptome profiling revealed the co-enrichment of G9a and H3K9me2 of multiple genes involved in the negative regulation of the WNT signaling pathway, in repression of EMT and extracellular matrix organization, leading to their repression in CRC [148]. G9a also methylates two non-histone substrates involved in CRC cell proliferation, FOXO1 (Forkhead family transcription factor) and p53 [59,106]. FOXO1 is methylated by G9a on K273, increasing the interaction between FOXO1 and the E3 ligase SKP2. This event decreases FOXO1 protein stability and promotes cellular proliferation in colon cancer [59]. These authors also demonstrated that G9a protein expression is increased in human colon cancer patient tissue samples associated with a decrease in FOXO1 protein level [59]. Likewise, G9a-mediated p53 dimethylation at lysine 373 was shown to increase Plk1 expression and consequently CRC cell growth [106]. 

#### 5.1.6. Hepatocellular Carcinoma

In hepatocellular carcinoma (HCC), targeting G9a is suggested as a novel therapy for HCC treatment as it drives tumorigenesis and aggressiveness (Table 3) [149,150]. Indeed, G9a is upregulated in HCC, which leads to the epigenetic silencing of the retinoic acid receptor responder protein 3 (RARRES3) tumor suppressor gene, thus triggering HCC proliferation and metastasis in vitro and in vivo [116]. Moreover, G9a was shown to enhance metastasis formation through an epigenetic regulation of EMT, as it interacts with SNAIL2 and HDACs at the E-cadherin promoter in order to inhibit E-cadherin transcription [151]. A recent study showed that G9a contributes to HCC initiation by escaping p53-induced apoptosis in DNA-damaged hepatocytes via the repression of Bcl-G expression, a pro-apoptotic Bcl-2 family member [152]. 

#### 5.1.7. Urinary Bladder Cancer

G9a was reported to be upregulated or amplified in urinary bladder cancer (UBC) [153]. G9a represents a promising therapeutic target for UBC as various G9a inhibitors decrease cell proliferation and increase cell death through the endoplasmic reticulum stress pathway [153]. Likewise, targeting G9a and DNMT methyltransferase activity with a novel dual inhibitor called CM-272 induces cell apoptosis and immunogenic cell death [153]. 

#### 5.1.8. Hematological Cancers

G9a is upregulated in hematological malignancies, for which G9a inhibitors have been identified as promising targets for patient management (Table 3) [154,155,156,157,158]. In T-lymphoblastic leukemia cells (T-ALL), inhibiting G9a activity suppresses cellular proliferation and induces apoptosis by downregulating the expression of Bcl-2 and upregulating the expression of Bax and caspase-3 [155]. Likewise, in chronic lymphocytic leukemia, targeting G9a and GLP was shown to stimulate cancer cell death [154]. In multiple myeloma, G9a fosters ReIB-dependent cancer growth and survival, whereas its depletion reduces the expression of ReIB and increases the expression of pro-apoptotic genes, such as Bim and BMF [118]. In acute myeloid leukemia (AML), G9a inhibition attenuates the transcriptional activity of the leukemogenic transcription factor HoxA9 and thus promotes AML proliferation, progression, and self-renewal [157]. In childhood acute lymphoblastic leukemia, G9a is reported to enhance the ability of cancer cells to migrate [159]. 

#### 5.1.9. Other Cancers

G9a represents an intriguing target in various other types of cancers (Table 3). In medulloblastoma, G9a drives H3K9me1/2/3 at the promoter of ubiquitin-specific protease 37 (USP37) to repress its gene expression [163]. USP37 controls cell proliferation by regulating the stability of the cyclin-dependent kinase inhibitor 1B (CDKN1B/p27Kip1) in cell cycle. Thus, blocking G9a inhibits cellular proliferation and tumorigenic potential of medulloblastoma cells [163]. Pre- or post-treatment of glioma cells with a G9a inhibitor sensitizes these cells to Temozolomide (TMZ), the first line therapy for glioblastoma patients, and increases its cytotoxicity [168]. Interestingly, authors demonstrated that the G9a inhibitor reprograms glioma cells and glioma stem-like cells to increase sensitivity to TMZ [164,168]. As previously described in breast cancer, HCC, and lung cancer, G9a interacts with SNAIL in order to mediate repression of E-cadherin and EMT in head and neck squamous cell carcinoma (HNSCC) [166]. Additionally, G9a was associated with cholangiocarcinoma, a highly malignant epithelial tumor of the biliary tree, where G9a-mediated H3K9 methylation suppressed the expression of the tumor suppressor gene LATS2, leading to the subsequent activation of the oncogenic YAP signaling pathway [120]. Recently in melanoma, elevated G9a levels promoted cancer progression through the activation of the WNT/β-catenin signaling by epigenetic silencing of the WNT antagonist DKK1 gene [165], or through the upregulation of the Notch1 signaling pathway, that further stimulates PI3K/AKT pathway [119]. 

### 5.2. G9a Tumor Suppressive Role

In stark contrast to its oncogenic roles, several studies demonstrated that G9a also promotes tumor suppressive functions. For example, G9a depletion increased the aggressiveness of lung tumor propagating cells (TPC) and accelerated disease progression and metastasis [167]. Inhibition of G9a derepresses genes that regulate the extracellular matrix. Patients with high levels of G9a displayed a better survival in early-stage lung cancer [167]. Interestingly, in glioblastoma, G9a inhibited HIF-1α-mediated migration via the methylation of the alpha subunit at lysine 674 [63]. 

## 6. Outlook

Over the last three decades since G9a was discovered, extensive studies were conducted to gain further insight into its physiological and pathophysiological roles. Aside from its key role in epigenetic repression through H3K9 methylation, G9a displays many biological functions, notably in gene expression, associated with its methylation of histone and non-histone substrates. Furthermore, a growing body of evidence indicates that G9a acts as a coregulator of transcription factors and steroid receptors, and could hence endorse other functions through these properties. Owing to its broad implication in biological activities, dysregulation of G9a expression is common to many types of cancers, and, as such, G9a represents a promising target for anti-cancer agents. Indeed, many inhibitors of G9a inhibitors have been synthetized and characterized, and could represent interesting therapeutic agents. 

## Figures and Tables

**Figure 1 life-11-01082-f001:**
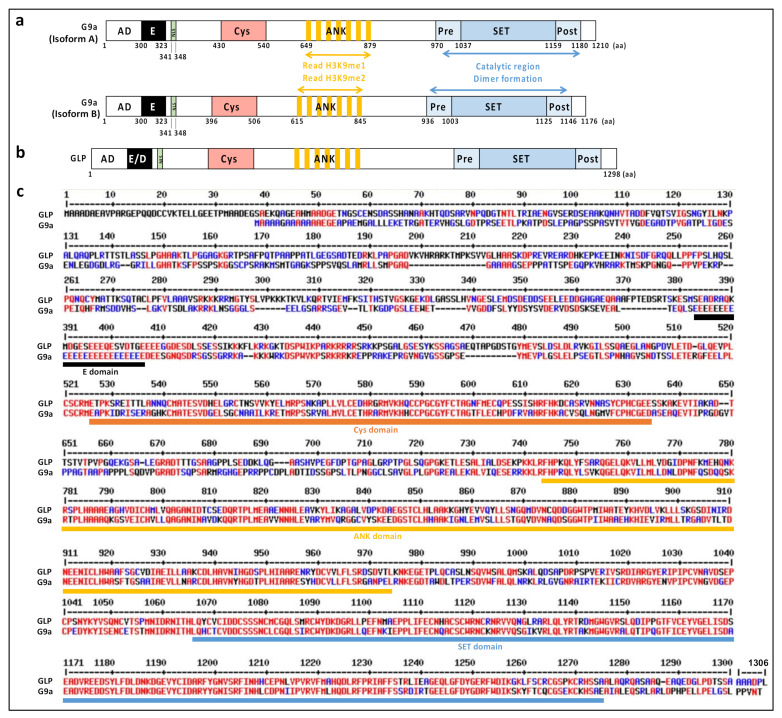
Schematic representation of the structure and domains of human G9a (**a**) and GLP (**b**). G9a and GLP contain different domains: an activation domain (AD), a Cys-rich region (Cys), an ankyrin repeat domain (ANK), and a SET domain composed of a core SET domain associated with pre- and a post-SET domains. G9a and GLP contain a nuclear localization signal (NLS). G9a also contains a Glu-rich region (E) and GLP a Glu/Asp-rich region (E/D). (**c**) Sequence alignment of G9a (NP_006700.3) and GLP (NP_079033.4). The alignment was performed using the MultAlin program [26] (http://multalin.toulouse.inra.fr/multalin) (accessed on 7 October 2021). Amino acids with 100% and >60% conservation are shown in red and blue, respectively.

**Figure 2 life-11-01082-f002:**
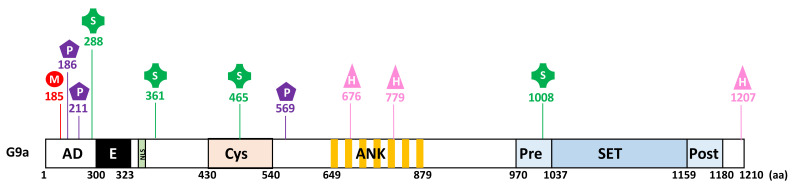
G9a undergoes several post-translational modifications including methylation (M), phosphorylation (P), sumoylation (S), and hydroxylation (H). The numbers indicate amino acid (aa) residues.

**Figure 3 life-11-01082-f003:**
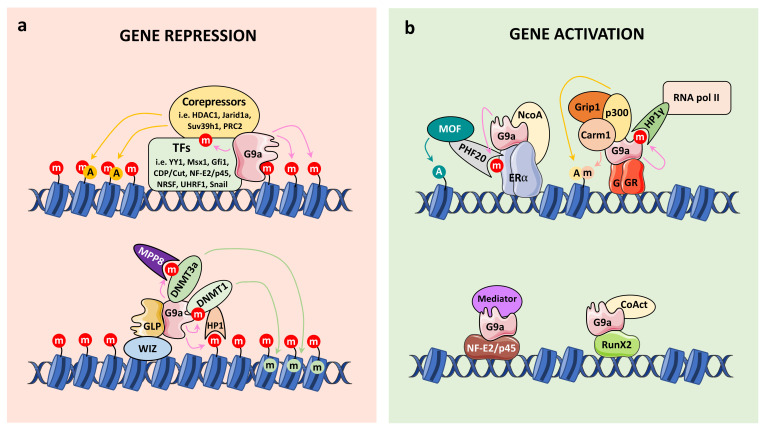
G9a acts as a transcriptional coregulator, either as a corepressor (**a**) or coactivator (**b**). (**a**) After G9a recruitment by some transcription factors (TFs), G9a methylates histones (red circles) leading to chromatin remodeling and gene repression. G9a also recruits corepressor proteins (i.e., other PKMTs and chromatin remodelers) and DNA methyltransferases (i.e., DNMT3a and DNMT1) in order to fully repress transcription via histone modifications (i.e., acetylation (orange circles) and DNA methylation (green circles)). Of note, G9a also methylates some TFs and DNA methyltransferases modulating their functions. (**b**) Conversely, G9a recruitment by the glucocorticoid receptor (GR), estrogen receptor (ERα), RunX2 and NF-E2/p45 leads to gene activation through the recruitment of specific coactivators (CoAct) (i.e., histone acetyltransferases and methyltransferases) and the transcription machinery (i.e., Mediator complex or RNA polymerase II).

**Figure 4 life-11-01082-f004:**
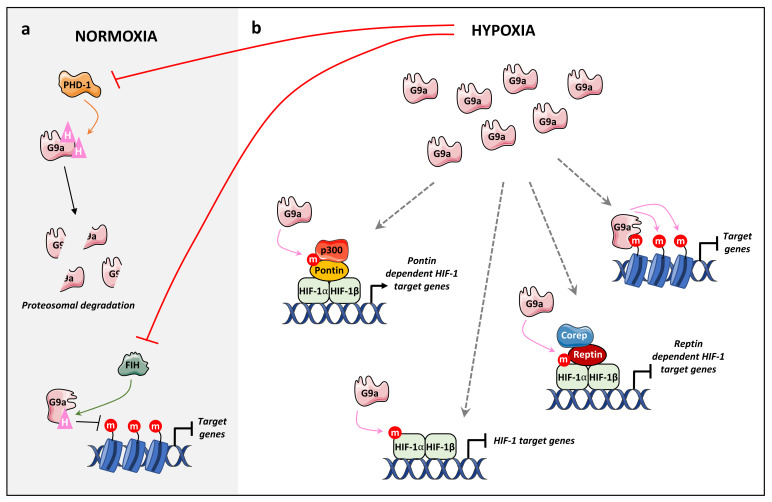
The role of G9a in hypoxia. (**a**) In normoxia, PHD-1 hydroxylates G9a on P676 and P1207 leading to proteosomal degradation. Likewise, FIH hydroxylates G9a on N779 impairing its ability to bind to H3K9me1/2 products. These hydroxylation processes are inhibited under hypoxic conditions resulting in an increase in the global level of G9a protein and H3K9me2. (**b**) In addition, under hypoxia, G9a methylates histones and non-histone targets. Hypoxia increases G9a-dependent H3K9me2 at the promoter regions of several genes leading to their repression. In addition, G9a methylates HIF-1α coregulators Pontin and Reptin during hypoxic stress, leading to the activation and repression of HIF-1α target genes. Finally, HIF-1 methylation by G9a suppresses HIF-1α transcriptional activity under hypoxia.

**Table 1 life-11-01082-t001:** List of histone substrates of G9a and their biological outcome. Nd: not determined.

Histone Types	Sites	Biological Outcome	References
Histone H3	H3K9me1	Transcriptional repression Heterochromatin formation	[5,25,37]
	H3K9me2		
	H3K9me3		
Histone H3	H3K27me1	Transcriptional repression Heterochromatin formation	[46,48]
Histone H3	H3K56me1	DNA replication	[49]
Histone H1.2	H1.2K187me	nd	[51]
Histone H1.4	H1.4K26me1	Transcriptional repression Chromatin structure	[50]
	H1.4K26me2		

**Table 2 life-11-01082-t002:** List of substrates of G9a categorized by their biological functions. Nd: not determined.

Functions	Substrates	Site	Biological Outcome	References
Transcription Factors	C/EBPb	K39	Inhibits transcriptional activity by repressing C/EBPb transactivation	[54]
	MyoD	K104me1/2	Inhibits MyoD transcriptional activity	[55]
	MEF2D	K267me1/2	Inhibits MEF2D transcriptional activity by preventing its recruitment on chromatin	[56]
	p53	K373me2	Inhibits transcriptional activity and p53-dependent apoptosis	[57]
	ERα	K235me2	Induces transcriptional activity by recruiting the PHF20/MOF HAT complex	[58]
	Foxo1	K273me1/2	Induces Foxo1 degradation	[59]
	KLF12	K313	nd	[36]
Chromatin remodeling factors and coregulators	G9a	K185me2/3	Induces specific glucocorticoid receptor transcriptional activity by recruiting HP1γ	[32,34,35]
	GLP	K205me2	Induces specific glucocorticoid receptor transcriptional activity by recruiting HP1γ	[32]
	Sirt1	K662	nd	[60]
	Pontin	K265, K267, K268, K274, K281, K285	Induces HIF-1 transcriptional activity by enhancing p300 recruitment	[61]
	Reptin	K67me1	Inhibits HIF transcriptional activity by recruiting corepressors	[62]
	HDAC1	K432	nd	[36]
	HIFα	K674me1/2	Inhibits HIF-1 transcriptional activity	[63]
	CSB	K170, K297, K448, K1054	nd	[36]
	MTA1	K532me1	Inhibits transcription by recruiting the assembly of the NuRD repressive complex	[64]
	ATF7IP (hAM)	K16me3	Induces transgene silencing by recruiting MPP8	[65]
Chromatin binding protein	CDYL1	K135me3	Decreases its interaction with H3K9me3	[36]
	WIZ	K305me3	nd	[36]
DNA methyltransferases	DNMT1	K70me2	nd	[36]
	DNMT3	K47me2	Inhibits transcription by recruiting MPP8/DNMT3/G9a/GLP repressive complex	[66]
Others	Acinus	K654me2	nd	[36]
	MDC1	K45me2	Induces ATM accumulation on damage sites	[67]
	Plk1	K209me1	Antagonizes T210 phosphorylation to inhibit Plk1 activity on DNA replication	[68]
	Lig1	K126me2/3	Maintenance in DNA methylation by promoting UHRF1 recruitment to replication foci	[69]

**Table 3 life-11-01082-t003:** Role of G9a in Cancer Biology.

G9a Roles	Cancer Types	G9a Biological Roles	References
Oncogenic	Breast Cancer	Suppresses tumor suppressor genes Enhances EMT Disrupts iron homeostasis Inhibits autophagy	[44,130] [123,131] [109] [132]
	Gastric Cancer	Suppresses tumor suppressor genes Inhibits apoptosis and autophagy Promotes metastasis	[138] [125,135,136,137] [139]
	Ovarian Cancer	Promotes metastasis Suppresses tumor suppressor genes Maintains PARP-inhibitor resistance	[111] [45,111] [140]
	Cervical Cancer	Induces angiogenesis Enhances tissue invasion	[142] [112]
	Endometrial Cancer	Enhances tissue invasion	[113]
	Prostate Cancer	Stimulates proliferation	[114]
	Lung Cancer	Enhances EMT Activates WNT signaling pathway Maintains lung cancer stemness Supports resistance to radiotherapy	[124,144,145,160] [121] [147] [161]
	Colorectal Cancer	Stimulates proliferation Enhances self-renewal and stemness Promotes resistance to chemotherapy	[59,106] [148] [162]
	Liver Cancer	Suppresses tumor suppressor genes Enhances EMT Inhibits cell apoptosis	[116] [151] [152]
	Bladder Cancer	Inhibits cell apoptosis and autophagy	[122,153]
	Brain Cancer	Stimulates proliferation Inhibits autophagy	[117,163] [164]
	Hematological malignancies	Enhances self-renewal and stemness Promotes migration Inhibits apoptosis and stimulates proliferation	[157] [159] [118,155]
	Skin Cancer	Promotes progression	[119,165]
	Head and Neck Cancer	Enhances EMT	[166]
	Bile duct Cancer	Suppresses tumor suppressor genes	[120]
Anti- oncogenic	Lung Cancer	Inhibits cancer progression	[167]
	Brain Cancer	Inhibits HIF-induced migration Inhibits cancer stemness	[63]

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
