# Peer review of "Structure, Activity, and Function of the Protein Lysine Methyltransferase G9a"

_life, 2021, doi:10.3390/life11101082_

Round 1

Reviewer 1 Report

This is a very mature manuscript from Poulard et al. describing the structure and role of the protein lysine methyltransferases G9a and GLP. This review is very comprehensive, with figures well supporting the text. Poulard et al. found a more unique take in this review, starting from the structure and biochemical features of G9a before moving towards its diverse functions and implications in cancer.

Only few specific remarks:

Section 2.1:

How is the conservation of G9a in other species? Human vs mouse? Are genes in the Su(var)3-9 family very similar? This could be very briefly mentioned. 

What is the functional relevance of isoform a/b? Which isoform is mainly expressed? 

Small mistakes/typos:

Introduction:

Abbreviation of lysine methyltransferases (PKMTs) -> add "Protein" lysine methyltransferases.

AdoMet: write abbreviation once in full 's-adenosyl-l-methionine' 

Figure 1B: "NLS" is missing.

2.2 "G9a and GLP preferentially bind to mono- and dimethylated H3K9, respectively" -> Is this correct? Please review.

3.1 typo "catalytic activity OF G9a"

4.1.2 has GR been abbreviated in the text body or only in the figures? 

4.2.1 typo "were reported to be increase"; typo "RunX2"

Legend figure 3: Starting with "(a) Under hypoxic condition" is not very intuitive. 

5. "In colorectal cancer, high levels of G9a in CRC"
"HDCAs" -> HDACs?

"In Acute myeloid leukemia (AML), G9a attenuates the transcriptional activity of HOXA9" -> Something is wrong here, should it be G9a INHIBITION attenuates? 

Reviewer 2 Report

In this review entitled “Structure, activity, and function of the protein lysine methyltransferase G9a”, Poulard et al., discuss structural features and biochemical features of G9aprotein lysine methyl transferase. The authors further discuss the various targets of G9a and its role in embryonic development, hypoxia, DNA repair and cancer.

While authors have discussed all the sections in detail and included most of the literature, which is appreciable, I find some major concerns as below:

  • The manuscript lacks organization in some sections, and it seems like an information dump. The authors should reorganize it, especially section 4 for the ease of read.
    1. The section 4.1.1 is confusing, and authors should modify it for clarity.
    2. The section 4.2.1 is very lengthy and can be compacted.
    3. The section 4.2.2 (labelled again as 4.2.1) is also very long. In its first paragraph, authors only discussed three references. The authors should write the summary of these studies and essence of their work rather than discussing their studies in detail.
    4. The authors should also write/propose the significance of G9a involvement in various processes/environment after each section.
  • For section 2, the authors should provide sequence and structural alignments for G9a and GLP for ease of understanding.
  • In section 3.3.1, authors mentioned that “phosphorylation of G9a on S211 does not change its methyltransferase activity”. What about S569?
  • Section 5 should be divided in small sections with each cancer type discussed in separate section.
  • The writing style can be improved and many grammatical mistakes throughout the manuscript should be corrected. For example, many sentences are either redundant, or can be combined. I am pointing out some errors below but there are many more:
    1. Page 4, paragraph 1- Alanine is hydrophobic and not hydrophilic.
    2. Page 4, section 3.3.1- “Similarly to most proteins, G9a is subjected by many PTMs that regulate its ability to bind new partners and impact its cellular functions” should be replaced with “Similar to most proteins, G9a is subjected to many PTMs that regulate its ability to bind …..”
    3. Section 3.3.2 First line- What is the difference between depletion and decrease in expression? Isn’t it the same?
    4. Page 5, First line, redundant information.
    5. There are many other lines which have redundant information or words, and the authors should go through each of these carefully.
    6. In many places references are missing (for example page 7 line 2).
    7. Figure legends for figure 2 should be improved.
    8. The authors should limit the use of furthermore and moreover in the manuscript.

Round 2

Reviewer 2 Report

1) While authors modified the manuscript significantly as suggested, I still feel the writing style should be improved more. Authors have only improved it where suggested and in some places the changes were not appropriate. Please modify the writing style of the manuscript and rearrange sentences for clarification.

2) The authors included only sequence alignment but not  structural  alignment of G9a and GLP.

3) References are lacking in some places

Author Response

We are grateful to reviewer #2 for his/her meticulous reading of our manuscript and for pointing out further concerns with our latest version. We have attempted to address these concerns as much as possible and hope that he/she now deems this new version suitable for publication.

1) While authors modified the manuscript significantly as suggested, I still feel the writing style should be improved more. Authors have only improved it where suggested and in some places the changes were not appropriate. Please modify the writing style of the manuscript and rearrange sentences for clarification.

We thank the referee for noting the significant effort made to address his/her first round of comments. We acknowledge that these comments improved the flow of our manuscript and were therefore surprised by this second round of comments on the writing style. However, the manuscript was once again proofread by a native English speaker, who has made several rearrangements and modifications where appropriate. We hope that you will be satisfied with this new version or that you may highlight the sections that remain amiss.

2) The authors included only sequence alignment but not structural alignment of G9a and GLP.

We have now added structural features on the alignment in Figure 1C.

3) References are lacking in some places

We have added some references where appropriate.